# Photothermal Combination Therapy for Metastatic Breast Cancer: A New Strategy and Future Perspectives

**DOI:** 10.3390/biomedicines13102558

**Published:** 2025-10-20

**Authors:** Zun Wang, Ikram Hasan, Yinghe Zhang, Tingting Peng, Bing Guo

**Affiliations:** 1Department of Breast and Thyroid Surgery, Shenzhen Baoan Women’s and Children’s Hospital, Jinan University, Shenzhen 518000, China; wzttkl@163.com; 2School of Biomedical Engineering, Medical School, Shenzhen University, Shenzhen 518060, China; ikramhasanszu2021@126.com; 3Shenzhen Key Laboratory of Advanced Functional Carbon Materials Research and Comprehensive Application, Harbin Institute of Technology, Shenzhen 518055, China; caroline.w.2006@gmail.com; 4State Key Laboratory of Bioactive Molecules and Druggability Assessment, Guangdong Basic Research Center of Excellence for Natural Bioactive Molecules and Discovery of Innovative Drugs, College of Pharmacy, Jinan University, Guangzhou 511436, China; 5School of Science, Harbin Institute of Technology, Shenzhen 518055, China

**Keywords:** metastatic breast cancer, PTT mechanism, photothermal therapy, combination therapy, limitation

## Abstract

Metastatic breast cancer (MBC) remains one of the most aggressive and fatal malignancies in women, primarily due to tumor heterogeneity, multidrug resistance, and the limitations of conventional therapeutic approaches. **Aim:** This review aims to evaluate recent advances in nanomaterial-based photothermal therapy (PTT) platforms and their potential in the treatment of metastatic breast cancer. **Method:** A comprehensive analysis of current literature was conducted to examine how various nanomaterials are engineered for targeted PTT, with particular emphasis on their mechanisms of action, synergistic applications with chemotherapy, immunotherapy, and photodynamic therapy, as well as their capacity to overcome challenges associated with targeting metastatic niches. **Results:** The findings indicate that nanotechnology-enabled PTT provides spatiotemporal precision, efficient tumor ablation, and reduced systemic toxicity, while significantly enhancing therapeutic outcomes when integrated into multimodal treatment strategies. Recent preclinical studies and early clinical trials further underscore advancements in imaging guidance, thermal efficiency, and site-specific drug delivery; however, issues related to biocompatibility, safety, and large-scale clinical translation remain unresolved. **Conclusions:** Nanomaterial-assisted PTT holds substantial promise for improving therapeutic efficacy against metastatic breast cancer. Future research should prioritize optimizing imaging resolution, minimizing adverse effects, and addressing translational challenges to accelerate clinical integration and ultimately enhance health outcomes for women.

## 1. Introduction

Breast cancer remains the most frequently diagnosed malignancy and a leading cause of cancer-related mortality among women worldwide. MBC represents the most advanced and life-threatening stage, characterized by the spread of malignant cells to distant organs, which accounts for the majority of breast cancer–related deaths. Current therapeutic regimens, including systemic chemotherapy, endocrine therapy, HER2-targeted therapy, and immunotherapy, often exhibit limited efficacy due to adverse side effects, tumor heterogeneity, and the emergence of multidrug resistance. PTT has recently emerged as a minimally invasive and highly specific modality for managing MBC [1,2,3,4]. This technique utilizes exogenous or endogenous photothermal agents capable of converting near-infrared (NIR) light into localized heat upon laser irradiation. The resultant hyperthermia induces irreversible cancer cell damage, triggers immunogenic cell death, and enhances drug penetration and immune cell infiltration. Due to its spatiotemporal precision, PTT enables localized ablation of tumor tissues while minimizing injury to surrounding healthy cells. Moreover, combinatorial therapeutic approaches integrating PTT with chemotherapy, immunotherapy, gene therapy, or imaging modalities have demonstrated synergistic therapeutic effects in preclinical models of breast cancer metastasis [2,4,5,6].

Metastasis is a multifaceted and dynamic process involving the detachment of tumor cells from the primary lesion, local tissue invasion, intravasation into the bloodstream or lymphatic vessels, and subsequent colonization of distant organs. This cascade is initiated by epithelial–mesenchymal transition (EMT), which endows tumor cells with migratory and invasive capacities. Once in circulation, cancer cells evade immune surveillance and establish pre-metastatic niches through interactions with the local microenvironment at secondary sites [7,8,9,10]. Key molecular mechanisms underlying this process include dysregulated signaling pathways, oncogene overexpression, and the inactivation of tumor suppressor genes. Collectively, these biological alterations contribute to therapeutic resistance and poor clinical outcomes. Globally, MBC poses a significant public health burden, with more than 2.3 million new cases reported annually. In the United States alone, approximately 168,000 women live with MBC, and the five-year survival rate remains around 29%, markedly lower than that of localized breast cancer. Common metastatic sites include the bone, lung, liver, and brain, with bone involvement occurring in nearly 75% of cases. Among molecular subtypes, HER2-positive and triple-negative breast cancers exhibit particularly aggressive metastatic behavior and poor prognosis, underscoring the persistent challenge of incurability despite recent advances in targeted and immune-based therapies [4,6,9,11].

Therapeutic failure in MBC remains a critical clinical challenge, driven by factors such as tumor heterogeneity, acquired drug resistance, and the adaptive capabilities of cancer cells within distant microenvironments. As metastases evolve, cancer cells accumulate genetic and epigenetic alterations that render them less susceptible to conventional therapies. The tumor microenvironment (TME) at metastatic sites offers a protective niche that hinders immune cell infiltration and limits drug diffusion [12,13,14,15,16]. Repeated treatment cycles often select for resistant cellular subpopulations, further compromising therapeutic outcomes. In this context, PTT offers a promising strategy for eradicating metastatic lesions by exploiting photoabsorbing nanomaterials that efficiently convert NIR light into localized hyperthermia. This precision-based thermal ablation minimizes collateral damage to healthy tissues and has proven particularly advantageous for treating metastases in organs such as the bone, lung, and liver (Figure 1), where conventional therapies frequently fail. Functionalization of nanoparticles with tumor-targeting ligands enhances their accumulation within metastatic niches, while the localized heating effect of PTT can remodel the tumor microenvironment and promote antitumor immune activation. Consequently, combining PTT with immunotherapy may yield synergistic benefits by suppressing residual tumor cells and preventing further metastatic dissemination [13,14,17,18,19,20].

However, photothermal combination therapy (PTCT) has emerged as an innovative and clinically significant strategy for treating MBC, integrating localized photothermal heating with complementary therapeutic modalities such as chemotherapy, immunotherapy, or chemodynamic therapy. In contrast to conventional systemic treatments, which often suffer from high toxicity and limited selectivity, PTCT enables precise spatiotemporal regulation of tumor ablation while simultaneously enhancing drug delivery, stimulating immune responses, and suppressing metastatic dissemination. The development of advanced photothermal nanomaterials with strong NIR-II absorption, real-time imaging capabilities, and multifunctional therapeutic properties holds transformative potential to overcome drug resistance and improve survival outcomes. Collectively, these advances highlight PTCT as a promising direction for future clinical translation [18,20].

Despite substantial progress, several critical challenges hinder the clinical realization of photothermal combination therapy for MBC. These include limited translation from preclinical studies, an incomplete understanding of synergistic mechanisms, and inadequate assessment of long-term biosafety and biodistribution. Current research is predominantly confined to small-animal models, which insufficiently recapitulate the complexity of human tumor heterogeneity and metastatic progression. Furthermore, achieving optimal photothermal performance, such as deep-tissue penetration, precise thermal regulation, and effective immune modulation, remains an ongoing challenge. Future investigations should therefore prioritize the design of biocompatible, multifunctional nanoplatforms incorporating image-guided, stimuli-responsive, and immune-integrative functionalities. Comprehensive in vivo validation and well-structured clinical trials will be essential to ensure therapeutic efficacy, safety, and eventual clinical applicability of this promising multimodal treatment paradigm [20,21,22].

Moreover, several in-depth reviews have recently studied the role of PTT in metastatic breast cancer management. For example, one study discussed the progress of nanotechnology-enabled PTT for treating primary breast cancer and metastatic lesions. Furthermore, this review underscores significant progress in multifunctional PTT nanoplatforms for breast cancer while emphasizing persistent challenges such as limited laser penetration, biosafety issues, lack of standardized protocols, and difficulties in clinical translation, particularly regarding tumor subtype variability and integration with immunotherapy [21]. However, another study, presented a comprehensive review on multifunctional nanoplatforms for breast cancer bone metastasis, highlighting recent advances in nanoparticle-mediated drug delivery, targeting strategies, and therapeutic outcomes. In addition, this review also addresses the major challenges hindering clinical translation, including biosafety, targeting specificity, and treatment standardization [22]. On the one hand, one research group, provided an in-depth review on advances in lipid-based nanocarriers for the treatment of breast cancer metastasis, focusing on their potential to improve drug solubility, stability, and targeted delivery. This review also discusses the therapeutic benefits, challenges in clinical translation, and future perspectives of lipid-based nanocarriers in managing metastatic breast cancer [23]. On the other hand, another research group, reviewed the impact of nanomedicine in women’s metastatic breast cancer, emphasizing its role in improving drug delivery, enhancing therapeutic efficacy, and reducing systemic toxicity. This review also highlights emerging nanotechnology-based strategies for diagnosis, treatment, and personalized management of metastatic breast cancer [24].

This review presents a comprehensive analysis of PTT as a novel and rapidly evolving treatment strategy for MBC, emphasizing its underlying mechanisms, current therapeutic applications, and recent technological progress. It discusses the wide range of nanomaterials designed for efficient photothermal conversion and highlights targeted delivery approaches that facilitate precise localization of PTT agents within metastatic microenvironments. Furthermore, the review explores the development of multimodal therapeutic systems that integrate PTT with complementary treatment modalities, such as chemotherapy, immunotherapy, and other synergistic strategies, to enhance overall therapeutic outcomes. Major limitations, including restricted light penetration, non-specific nanoparticle accumulation, and heterogeneity in patient response, are critically evaluated. The discussion also outlines the current challenges and future directions required to advance PTT-based strategies from preclinical research to clinical implementation, underscoring the importance of improving imaging precision, optimizing thermal control, and minimizing adverse effects. By consolidating recent findings and identifying translational opportunities, this review aims to facilitate the clinical adaptation of PTT and promote improved therapeutic outcomes for women with metastatic breast cancer. The integration of nanotechnology and molecular targeting holds remarkable promise for reshaping the therapeutic landscape of metastatic disease and advancing precision oncology. This review systematically searched PubMed, Web of Science, Scopus, Embase, and Google Scholar (2010–2025) using the keywords “photothermal therapy”, “combination therapy”, “synergistic therapy”, and “metastatic breast cancer.” Only peer-reviewed English studies on photothermal combination therapies for metastatic breast cancer were included. Data on agents, therapeutic combinations, imaging, and outcomes were synthesized to highlight key strategies, efficacy, and future directions (Figure 2).

## 2. Pathophysiology of Metastatic Breast Cancer

Metastatic breast cancer (MBC), also known as Stage IV breast cancer, is the most advanced stage of the disease and remains the leading cause of mortality among breast cancer patients (Figure 1). The pathophysiology of MBC involves a complex biological cascade that allows primary tumor cells to spread and establish secondary tumors in distant organs. Moreover, the process begins with local invasion, where cancer cells detach from the primary tumor mass through the epithelial–mesenchymal transition (EMT). After EMT, tumor cells degrade the extracellular matrix and invade nearby blood or lymphatic vessels, forming protective emboli to enhance their survival and metastatic potential [1,25,26,27,28,29,30]. In addition, the circulating tumor cells (CTCs) then adhere to the endothelial lining of distant capillaries and extravasate into the parenchyma of distant organs, influenced by organ-specific cues and adhesion molecules. Before colonization, the primary tumor can influence distant sites by secreting factors that modify the local microenvironment, preparing a “pre-metastatic niche”. For sustained growth at the metastatic site, cancer cells induce angiogenesis through the secretion of pro-angiogenic factors like VEGF. However, MBC is characterized by significant intratumoral and intertumoral heterogeneity, contributing to drug resistance and treatment outcomes. Understanding the pathophysiological mechanisms of MBC is crucial for identifying new therapeutic targets and improving patient outcomes [26,27,28,31].

## 3. Mechanism of Photothermal Therapy

PTT and PDT are two light-based cancer treatments that differ fundamentally in their mechanisms. PTT relies on the non-radiative relaxation of photoexcited electrons in photothermal agents, such as gold nanorods, carbon-based nanomaterials, or semiconducting polymers, to generate localized heat, inducing protein denaturation, membrane disruption, and apoptosis or necrosis in an oxygen-independent manner, making it effective even in hypoxic tumors. In contrast, PDT depends on photoactivated photosensitizers transferring energy to molecular oxygen to produce reactive oxygen species (ROS), including singlet oxygen and free radicals, which cause oxidative damage to lipids, proteins, and DNA, resulting in apoptosis or necrosis; this process is oxygen-dependent, limiting its efficacy in hypoxic regions. While PTT’s cytotoxicity is primarily thermal, PDT’s is oxidative, and modern nanoplatforms often combine both approaches to exploit the synergistic effects of heat and ROS, enhancing therapeutic outcomes and overcoming the individual limitations of each modality [31,32,33].

PTT is a minimally invasive treatment method that uses light energy in the NIR region to induce localized hyperthermia for cancer cell ablation. PTAs, such as metal-based nanomaterials, carbon-based nanomaterials, semiconductors, and organic materials, absorb NIR light and dissipate the energy as heat, raising the local temperature of tumor tissues to cytotoxic levels. This leads to irreversible damage to cellular structures and ultimately, cancer cell death. The process involves electron-phonon and phonon-phonon interactions, which rapidly elevate the local temperature in the tumor microenvironment. Moreover, PTT can induce immunogenic cell death (ICD), which stimulates antigen-presenting cells (APCs) and activates cytotoxic T lymphocytes (CTLs). The NIR-I and NIR-II windows allow optimal tissue penetration, making it suitable for treating deep-seated or metastatic lesions [31,32,34,35,36]. When combined with systemic immune activation, PTT offers a compelling approach to tackling metastatic breast cancer. PTT for breast cancer operates through the conversion of light energy into heat by photothermal agents, such as nanoparticles or dyes, targeting tumor tissues. When these agents are irradiated with near-infrared (NIR) light, electrons in their molecules are excited from the ground singlet state (S_0_) to an excited singlet state (S_1_), as depicted in the Jablonski diagram. Rather than releasing this energy as light, the electrons predominantly undergo non-radiative relaxation, converting the absorbed energy into localized heat [36,37]. In addition, this heat raises the temperature of the tumor microenvironment, leading to protein denaturation, membrane damage, and subsequent apoptosis or necrosis of breast cancer cells. While some energy may contribute to fluorescence or reactive oxygen species generation in combination therapies, the primary mechanism in PTT is thermal ablation. Thus, the Jablonski diagram provides a clear illustration of how light absorption and energy relaxation at the molecular level translate into effective, localized cancer cell destruction (Figure 2) [31,33,34,35,37].

For PTT, it is crucial to describe temperature time profiles and quantitative thermal dose metrics, such as the cumulative equivalent minutes at 43 °C (CEM43), to ensure accurate evaluation of therapeutic efficacy and reproducibility. Detailed illumination parameters, including laser wavelength, irradiance (mW·cm^−2^), exposure duration, and total fluence (J·cm^−2^), should be fully described, along with spot size and beam uniformity. Real-time temperature monitoring using thermal cameras, fiber-optic probes, or infrared thermometry should capture spatial and temporal heating behavior, both in vitro and in vivo. Reporting thermal dose allows comparison across studies and distinguishes photothermal effects from other mechanisms, such as photodynamic or photochemical processes. Together, standardized thermal profiling and illumination documentation provide the mechanistic transparency necessary to interpret PTT outcomes and optimize therapeutic safety and precision [38,39].

To maintain mechanistic clarity in photodynamic therapy (PDT), it is essential to quantitatively describe reactive oxygen species (ROS) generation, particularly singlet oxygen (^1^O_2_), and assess oxygen dependence under well-defined irradiation conditions. Reliable assays such as Singlet Oxygen Sensor Green (SOSG), DPBF bleaching, EPR spin trapping, or direct near-infrared (1270 nm) phosphorescence should be used to confirm ^1^O_2_ formation, while general ROS indicators (e.g., DCFH-DA) can complement species-specific results. Experimental details, including light wavelength, irradiance, fluence, photosensitizer concentration, and oxygen levels (normoxia versus hypoxia) must be clearly documented. Comparative studies under different oxygen conditions and with ^1^O_2_ quenchers (e.g., sodium azide) help confirm oxygen-dependent mechanisms. Reporting quantitative parameters such as singlet oxygen quantum yield (ΦΔ), ROS generation rates, and oxygen consumption ensures reproducibility and strengthens mechanistic interpretation. These practices collectively enable accurate evaluation of PDT efficacy and its dependence on oxygen-mediated photochemical pathways [40,41].

However, preclinical studies using small-animal models provide controlled settings for testing optical imaging probes and therapies, offering insights and proof of concept. However, they may overestimate translational feasibility due to differences in tissue characteristics between animals and humans. Early-phase clinical evidence, while limited, is crucial for assessing safety and preliminary efficacy in humans. Key considerations include optical penetration differences between NIR-I and NIR-II wavelengths, anatomical accessibility issues for deep lesions, and complexities associated with interventional delivery methods. Aligning translational expectations with these factors can prevent overinterpretation of preclinical findings [38,39,40,41] (Table 1).

## 4. Targeting Metastatic Sites Using PTT

PTT is a promising strategy for metastatic breast cancer due to its precise spatiotemporal control and minimally invasive nature. PTT converts near-infrared light into localized heat using photothermal agents (PTAs), which induce hyperthermia-mediated cell death in tumors. In metastatic breast cancer, where cancer cells have spread from the primary tumor to distant organs, PTT provides a unique opportunity to target these secondary lesions selectively. However, by functionalizing nanoparticles with tumor-homing ligands, PTAs can accumulate specifically at metastatic sites through passive and active targeting mechanisms, minimizing off-target effects and preserving healthy surrounding structures [31,32,35,39,42,43]. Recent advances in nanotechnology have refined PTT for metastatic applications, with multifunctional nanoplatforms that combine imaging capabilities, targeting moieties, and therapeutic functions. Moreover, integrating PTT with imaging modalities allows for real-time tracking and assessment of therapeutic efficacy. Stimuli-responsive PTAs that respond to tumor-specific features can improve heat generation precision, providing an adaptive approach to treat metastatic sites with varied microenvironments. However, the future of PTT in managing metastatic breast cancer lies in developing clinically translatable, biocompatible, and multifunctional platforms capable of precise metastatic targeting with minimal toxicity [31,32,38,39,40].

However, PTT can be classified into monotherapy (PTT alone) and combination approaches that merge PTT with complementary treatments such as chemotherapy, immunotherapy, PDT, or radiotherapy to enhance therapeutic outcomes. PTT monotherapy induces localized hyperthermia through near-infrared (NIR) light–activated agents, leading to tumor cell apoptosis or necrosis at temperatures above 50–60 °C for 5–10 min in ablative protocols. Conversely, mild PTT operates at lower temperatures (41–45 °C for 5–20 min), triggering immunogenic cell death (ICD), heat shock protein (HSP) release, and T-cell activation, which contribute to systemic immune responses. When combined with chemotherapy, PTT enhances drug diffusion and tumor sensitivity, while integration with immunotherapy improves immune checkpoint response. PTT–PDT systems utilize reactive oxygen species (ROS) to achieve synergistic cytotoxic effects, and PTT–radiotherapy hybrids boost tumor oxygenation and radiosensitivity. Overall, these multimodal PTT strategies merge thermal and biological mechanisms to deliver improved precision, efficacy, and safety, representing a promising direction for the treatment of metastatic breast cancer [42,43,44,45,46,47] (Table 2).

### 4.1. Synthesis of Photothermal Agents

Developing targeted PTT systems for metastatic breast cancer requires the synthesis of effective agents with strong near-infrared absorption, high conversion efficiency, good biocompatibility, and potential for functionalization. Common nanomaterials include gold nanorods (GNRs), graphene oxide (GO), black phosphorus (BP), and copper sulfide (CuS) nanoparticles. However, gold nanorods are favored due to their tunable surface plasmon resonance, ease of synthesis, and biocompatibility. GNRs are grown using a seed-mediated growth method, with the aspect ratio determining their optical absorption peak. In addition, the surface of these GNRs is modified with polyethylene glycol (PEG) to improve circulation time and reduce toxicity. Copper sulfide nanoparticles are suitable for deep tissue PTT due to their intrinsic NIR absorbance and photostability. However, black phosphorus nanosheets can be functionalized with PEG or targeting peptides and systematically delivered through intravenous injection [48,49,50]. Moreover, advances in nanomaterial engineering continue to improve targeting efficiency, reduce off-target toxicity, and offer multifunctionality for combined therapy and imaging.

One study presents a novel approach to augment breast cancer chemotherapy by combining standard chemotherapeutic agents with gold nanoparticles irradiated by visible light, effectively harnessing photothermal energy to enhance treatment efficacy. In their experimental design, they load conventional chemotherapy drugs onto gold nanoparticles, then expose them to visible-light irradiation, which triggers localized heating around tumor cells. This mild hyperthermia potentiates chemotherapy’s cytotoxic effects, promoting more efficient tumor cell killing compared to drug treatment alone. In addition, the visible light-induced photothermal effect amplifies drug uptake and accelerates apoptotic pathways, thereby overcoming limitations like drug resistance and poor drug penetration. Importantly, the study demonstrates this synergistic impact both in vitro, showing heightened cancer cell mortality, and in vivo via mouse models, where the combined therapy leads to significantly greater tumor suppression. By strategically merging photothermal and chemotherapeutic mechanisms into a unified nanoplatform, the research highlights a promising strategy for improving breast cancer treatment outcomes [51,52,53]. However, in a novel biomimetic strategy, the researchers developed “nanoplatelets” by cloaking polymeric nanoparticles with co-encapsulated doxorubicin (DOX) and the FDA-approved photothermal agent indocyanine green (ICG) in a natural platelet membrane (PM). In addition, this design enabled the nanoparticles to evade immune clearance while leveraging the platelet’s inherent affinity for tumor-associated markers: the PM’s P-selectin binds with the cancer cell surface receptor CD44, allowing the nanoplatelets to specifically capture and clear circulating tumor cells (CTCs) in both blood and lymph systems, which are key conduits for metastasis. However, compared with uncoated counterparts, these PM-coated NPs exhibited enhanced uptake by MDA-MB-231 breast cancer cells and significantly greater cytotoxicity under combined chemo-photothermal activation. In vivo, in multiple breast tumor mouse models, including xenograft and orthotopic systems, the nanoplatelets not only completely ablated primary tumors but also effectively prevented metastatic spread. In addition, the study positions platelet membrane–coated nanoplatelets as a highly promising platform for actively targeting and eliminating CTCs, thereby offering a compelling avenue for inhibiting breast cancer metastasis via synergistic chemo-photothermal therapy [51].

In another study, researchers developed a nanotherapeutic platform in which gold nanorods (GNRs) were wrapped with DNA and loaded with doxorubicin (GNR@DOX) to synergistically combine chemotherapy and photothermal ablation for metastatic breast cancer. Moreover, in vitro, exposure to near-infrared (NIR) light triggered both localized heating and controlled DOX release, resulting in significantly greater cytotoxicity against 4T1 breast cancer cells compared to free DOX. In an orthotopic 4T1 murine model, systemic administration of GNR@DOX followed by NIR irradiation not only suppressed primary tumor growth but also effectively reduced lung metastases. In addition, histological (H&E) and immunohistochemical analyses confirmed that tumor inhibition and anti-metastatic effects were facilitated by induction of apoptosis or necrosis and disruption of tumor microvasculature. These findings demonstrate the strong potential of GNR@DOX as an integrated chemo-photothermal nanoplatform for treating metastatic breast cancer [53]. However, in one study, macrophage-based biohybrid microrobots (IDN@MC) were engineered by integrating macrophages with metal–organic frameworks (MOFs) to achieve breast cancer photothermal immunotherapy through pyroptosis induction. These multifunctional microrobots were constructed using a novel indocyanine green (IR) derivative as a fluorescent photosensitizer and DAC-loaded ZIF-8 nanoparticles (DZNPs) as therapeutic carriers. The macrophages served as both autonomous motors and drug reservoirs, leveraging their innate tumor-homing ability for targeted accumulation and precise delivery within the tumor microenvironment. Moreover, encapsulation of DZNPs within macrophages enabled pH-responsive, sustained DAC release, which facilitated temporal regulation of macrophage repolarization toward the M1 phenotype. Coupled with the IR photosensitizer, the microrobots offered fluorescence-guided navigation and controllable photothermal activity. Once localized at the tumor site, the gradual degradation of DZNPs promoted macrophage polarization to M1, enhancing tumor recognition and phagocytosis, while simultaneously transferring IR and DAC directly into tumor cells. Subsequent near-infrared laser irradiation induced localized hyperthermia, which, in concert with chemotherapy, triggered pyroptotic cell death. In addition, the pyroptotic tumor cells released inflammatory signals that stimulated systemic antitumor immunity, thereby suppressing primary tumor growth and metastatic progression. Collectively, this integrated platform highlights macrophage-based microrobots as a promising strategy for synergistic breast cancer therapy and metastasis prevention (Figure 3) [52].

### 4.2. Surface Functionalization for Targeting

Surface functionalization of photothermal agents is a crucial step in enhancing the selectivity and therapeutic efficiency of PTT for MBC. Functionalization strategies aim to endow nanoparticles with active targeting capabilities, improve biocompatibility, prolong systemic circulation, and facilitate accumulation at metastatic sites. Typically, photothermal nanomaterials such as gold nanorods, black phosphorus nanosheets, or copper sulfide nanoparticles are first coated with polyethylene glycol (PEG) to improve colloidal stability and reduce recognition by the mononuclear phagocyte system. Subsequently, specific targeting ligands, such as monoclonal antibodies (e.g., trastuzumab for HER2+ tumors), peptides (e.g., RGD for integrin αvβ3), or small molecules (e.g., alendronate for bone targeting), are conjugated to the nanoparticle surface to enable receptor-mediated binding to metastatic breast cancer cells or their microenvironment [54,55].

In one study, researchers engineered a multifunctional nanotheranostic platform composed of peptide-targeted gold nanocages loaded with FePt nanoparticles, designated as iRGD-PEG/AuNCs@FePt (Au@FePt NPs), to integrate both imaging and therapy for breast cancer. These nanoparticles enabled dual-modal imaging via photoacoustic imaging (PAI) and magnetic resonance imaging (MRI), ensuring precise tumor localization. Under mild PTT and X-ray-induced dynamic therapy (XDT), the Au@FePt NPs generated elevated levels of ROS and local heat, amplifying Fenton-like reactions through released Fe^2+^ and accelerating ferroptosis coupled with apoptosis in tumor cells. Furthermore, the synergistic effect of mild hyperthermia and radiation effectively potentiated ferroptotic cell death, thereby achieving enhanced tumor ablation while being guided and monitored noninvasively via MRI and PAI. This all-in-one, imaging-guided approach demonstrates a promising strategy for precise, effective antitumor therapy by combining photothermal, radiation, and ferroptosis-inducing mechanisms in a single nanoplatform [55].

However, one study presents a novel DSP-Zn@PEG-ALN nanoparticle system, comprising a cisplatin prodrug (DSP) coordinated within a zinc-based coordination polymer core, coated with PEG, and functionalized with the bone-seeking bisphosphonate alendronate (ALN). In addition, engineered to be ~55 nm in size, these nanoparticles effectively extravasate through the ≈80 nm bone capillary clefts and preferentially accumulate in metastatic bone lesions. In vitro, they exhibit significantly enhanced affinity for hydroxyapatite, and in vivo biodistribution shows about four times more platinum delivered to metastatic bone compared to healthy bone tissue. Importantly, compared to non-targeted controls and free cisplatin, the DSP-Zn@PEG-ALN NPs more effectively inhibited tumor growth and suppressed osteoclastic bone destruction, all while reducing the systemic toxicity of cisplatin, demonstrating a promising strategy for targeted chemotherapy against bone metastatic breast cancer [56].

Another research group developed cancer cell membrane–coated poly(lactic-co-glycolic acid) (PLGA) nanospheres (M@P-WIs) with multiple functional advantages. The homologous membrane camouflage enables precise tumor-targeted delivery, while the co-loading of the photothermal dye IR-780 and the CAF-modulating agent WRG-28 provides near-infrared (NIR)-triggered, spatiotemporal control over stromal regulation. Moreover, this design establishes a sequential therapeutic strategy of “tumor eradication followed by stromal remodeling.” In practice, M@P-WIs first attach to tumor cells, where NIR irradiation induces photothermal ablation and immunogenic cell death (ICD). Subsequently, controlled release of WRG-28 suppresses metastasis and progressively disrupts the extracellular matrix (ECM) barrier. At the same time, photothermally liberated tumor-associated antigens stimulate systemic antitumor immune responses. Through the combined actions of photothermal ablation, stromal modulation, and immune activation, M@P-WIs offer an innovative therapeutic approach that avoids the risks of tissue imbalance associated with complete CAF depletion, presenting a promising platform for breast cancer treatment (Figure 4) [57]. However, one study, reports the design and in vitro evaluation of novel hybrid iron oxide gold nanoparticles (Fe_3_O_4_–Au NPs) functionalized with a tumor-penetrating Hsp70 peptide (TPP) via a PEG_4_ linker, termed TPP-PEG_4_-FeAuNPs, which selectively target membrane-bound Hsp70 (mHsp70) overexpressed on aggressive triple-negative breast cancer (TNBC) cells such as 4T1 and MDA-MB-231, while sparing normal cells. In addition, these nanoparticles serve a dual function: the iron-oxide core enables MRI contrast, while the gold shell enhances radiosensitivity by releasing secondary (Auger) electrons under X-ray irradiation. After 24 h incubation, TPP-PEG_4_-FeAuNPs achieve significantly higher uptake in mHsp70-positive TNBC cells than non-conjugated or scrambled-peptide FeAuNPs, leading to induction of G2/M cell-cycle arrest, DNA double-strand breaks, and apoptotic cell death following irradiation. Crucially, addition of the ROS scavenger N-acetyl-L-cysteine (NAC) completely abolishes the radiosensitizing effect, strongly indicating that nanoparticle-induced reactive oxygen species (ROS) are the central mediators of apoptosis. The authors thus demonstrate a promising Hsp70-targeted nanoplatform that combines molecularly specific tumor targeting, imaging capability, and ROS-mediated radiosensitization to overcome radioresistance in TNBC cells [58].

### 4.3. NIR Irradiation and Heat Generation

NIR irradiation plays a pivotal role in PTT by enabling deep tissue penetration and localized heat generation at metastatic breast cancer (MBC) sites. The NIR window, typically between 700 and 1100 nm, allows minimal absorption and scattering by biological tissues, making it ideal for non-invasive external activation of photothermal agents accumulated in tumors. Upon exposure to NIR light, photothermal nanomaterials such as gold nanorods, black phosphorus nanosheets, or copper sulfide nanoparticles absorb the light energy and efficiently convert it into heat through nonradiative relaxation processes. This localized hyperthermia (typically reaching temperatures between 42 and 50 °C) induces apoptosis or necrosis in cancer cells while sparing surrounding healthy tissues [54,55,56,57,58]. For example, one study presents the design of a multifunctional nanoplatform that integrates imaging and therapy to overcome the limitations of conventional treatments. By employing second near-infrared (NIR-II) excitation, the system achieves deeper tissue penetration and higher spatial resolution, enabling precise imaging guidance for therapeutic interventions. In addition, the nanoplatform is engineered to deliver chemotherapeutic agents while simultaneously generating localized hyperthermia under NIR-II irradiation, which not only enhances drug release but also potentiates CDT through Fenton or Fenton-like reactions to produce toxic reactive oxygen species within the tumor microenvironment. This synergistic approach results in amplified tumor ablation, effective suppression of bone metastases, and reduced systemic toxicity. Overall, the work highlights a promising theranostic strategy that combines multimodal treatment with NIR-II guided imaging, offering significant potential for improving therapeutic efficacy and safety in managing breast cancer bone metastases (Figure 5) [59].

However, another study explores a promising nanoplatform for the treatment of bone metastases originating from breast cancer, a common and difficult-to-treat complication in advanced breast cancer patients. In this work, researchers developed a biodegradable and biocompatible nanoparticle system by encapsulating the near-infrared (NIR) photosensitizer IR780 within poly(lactic-co-glycolic acid) (PLGA) nanoparticles. These nanoparticles, referred to as IR780@PLGA, are designed to accumulate at tumor sites after systemic administration due to the enhanced permeability and retention (EPR) effect. Upon exposure to NIR laser irradiation (typically around 808 nm), the IR780 molecules efficiently convert absorbed light into heat, achieving localized hyperthermia that selectively destroys cancer cells without harming surrounding healthy tissues. Moreover, the photothermal conversion efficiency of IR780 within the PLGA matrix proved sufficient to raise the temperature to therapeutic levels capable of inducing tumor cell apoptosis. In in vivo models of breast cancer bone metastasis, mice treated with IR780@PLGA followed by NIR irradiation showed significant suppression of tumor growth in the bone, reduction in metastatic lesions, and preservation of bone architecture, compared to control groups. Furthermore, the treatment also demonstrated low systemic toxicity and good biocompatibility, suggesting its safety for potential clinical translation. Overall, this study highlights the effectiveness of an NIR-triggered, nanoparticle-mediated photothermal therapy approach for specifically targeting and treating breast cancer metastasis in bone, offering a minimally invasive and highly localized treatment option for advanced-stage breast cancer [60].

In another study, the authors first established, via bioinformatic analyses, that triple-negative breast cancer (TNBC) exhibits a significantly elevated ferroptosis potential index (FPI), derived from expression profiles of ferroptosis regulatory genes, in both cell lines and patient tumor tissues compared to non-TNBC, suggesting TNBC may be particularly susceptible to ferroptosis-based therapies. Furthermore, they then engineered biocompatible iron pyrite (FeS_2_) nanocrystals coated with hyperbranched PEG for use as “phototheranostic” agents: these nanocrystals efficiently absorb near-infrared (NIR) light, enabling high-contrast in vivo imaging via multispectral optoacoustic tomography (MSOT) and delivering potent photothermal effects with a conversion efficiency of about 63.1%. When irradiated with NIR light, the heat generated thermodynamically accelerates Fenton chemistry within tumors, simultaneously inducing ferroptosis (iron-dependent lipid peroxidation) and apoptosis in TNBC cells. Moreover, this dual-pathway cell death is mediated by modulation of key signaling networks including p53, FoxO, and HIF-1, as well as suppression of metabolic pathways related to glutathione and amino acid homeostasis. In preclinical models, this single nanoplatform both imaged and ablated primary and metastatic TNBC with minimal toxicity, offering a streamlined, image-guided therapeutic strategy to overcome the paucity of molecular targets and the aggressive metastatic behavior characteristic of TNBC. Importantly, NIR-induced hyperthermia not only causes direct tumor cell death but also modulates the tumor microenvironment by enhancing blood flow, increasing membrane permeability, and promoting immune cell infiltration, thereby amplifying therapeutic outcomes. The combination of NIR irradiation with precise nanomaterial design is thus critical to achieving targeted, effective, and safe PTT for metastatic breast cancer treatment [61].

### 4.4. Systemic Administration

Systemic administration, typically via intravenous injection, is the primary delivery route for photothermal agents aimed at targeting metastatic breast cancer (MBC) sites, enabling widespread biodistribution and the potential to reach disseminated tumor cells. Following administration, engineered nanoparticles leverage the enhanced permeability and retention (EPR) effect to passively accumulate in tumor tissues characterized by leaky vasculature and impaired lymphatic drainage. In addition, active targeting is further achieved by functionalizing nanoparticles with ligands such as antibodies, peptides, or small molecules that recognize receptors overexpressed on metastatic cancer cells or the tumor microenvironment, thereby increasing tumor-specific uptake and minimizing off-target effects [62,63,64]. Systemic delivery of photothermal agents for breast cancer metastasis necessitates overcoming multiple biological barriers, including rapid clearance by the reticuloendothelial system (RES), nonspecific distribution, and penetration of protective barriers such as the blood–brain barrier (BBB) in cases of brain metastases. However, to enhance tumor-targeting efficiency and prolong circulation time, nanoparticles are often functionalized with polyethylene glycol (PEG) or tumor-specific ligands, improving their stability and binding affinity to cancer cells. In addition, strategies incorporating stimuli-responsive coatings enable controlled release or activation of photothermal agents specifically at tumor sites, thereby maximizing therapeutic efficacy while reducing off-target effects. These advances facilitate effective systemic administration of PTT agents capable of treating disseminated disease [65,66,67,68].

However, combination therapies integrating PTT with chemotherapy, immunotherapy, or radiotherapy have shown synergistic effects against metastatic breast cancer in preclinical studies. Hyperthermia generated by PTT not only induces direct tumor cell death but also disrupts the tumor microenvironment, enhancing vascular permeability and drug delivery. Moreover, PTT-induced immunogenic cell death promotes the release of tumor antigens and danger signals that activate innate and adaptive immune responses. When combined with immune checkpoint inhibitors, systemic PTT can stimulate systemic antitumor immunity, offering a promising approach to control metastatic progression and reduce tumor recurrence. Such multimodal regimens are at the forefront of translational research aiming to improve patient outcomes in metastatic breast cancer [68,69,70]. For example, one study presents the design and evaluation of a novel theranostic nanoplatform, ICG/Fe_3_O_4_@PLGA-ZOL nanoparticles, engineered to treat breast cancer metastasis in the tibia via targeted photothermal therapy (PTT). These nanoparticles leverage dual targeting: zoledronate (ZOL) for bone affinity and superparamagnetic iron oxide (Fe_3_O_4_) for magnetic guidance. Both indocyanine green (ICG) and Fe_3_O_4_ serve as photothermal agents, converting near-infrared (NIR) light into heat. Under external magnetic field application, the nanoparticles accumulated in the proximal tibia’s medullary cavity. Upon NIR irradiation (1 W/cm^2^ for 5 min), they induced a rapid temperature rise (up to ~52 °C), significantly higher than less-targeted or non-targeted controls. In addition, this led to marked suppression of tumor progression, delayed leg swelling, reduced bone destruction, and extended survival in treated mice. Micro-CT and ultrasound imaging revealed preservation of bone volume and structure, while histological and TRAP assays confirmed reduced tumor burden, osteoclast activity, and bone resorption. Notably, the dual-targeted ICG/Fe_3_O_4_@PLGA-ZOL with PTT demonstrated superior therapeutic effects compared to single-target or non-targeted nanoparticle controls, highlighting its promise as an effective and precise treatment strategy for deep-seated breast cancer bone metastases [71].

## 5. Treatment of Breast Cancer Metastasis

Breast cancer metastasis is a significant challenge in oncology, often leading to a poor prognosis. Traditional treatments like chemotherapy, radiation, and surgery have limitations, such as systemic toxicity and incomplete tumor eradication. However, PTT has gained attention as a promising and minimally invasive approach for treating metastatic breast cancer. This technique utilizes nanomaterials activated by NIR light to generate localized heat, effectively destroying tumor cells with precision while minimizing damage to surrounding healthy tissue [72,73,74]. PTT relies on light-absorbing nanoparticles that accumulate in metastatic tumor sites and convert light into heat upon NIR laser irradiation. Furthermore, tumor-specific targeting can be achieved by functionalizing nanoparticles with tumor-targeting ligands, and immune system activation can be triggered by releasing tumor-associated antigens. Imaging-guided PTT for metastatic lesions requires precise imaging. PTT strategies can be tailored for specific metastatic sites, including lymph node metastases, lung metastases, brain metastases, liver metastases, and bone metastases. Combination therapies, such as immunotherapy, chemotherapy, and PTT, can further enhance the efficacy of PTT [75,76,77].

Photothermal combination therapy (PTCT) for metastatic breast cancer offers significant advantages by enhancing tumor selectivity, minimizing systemic toxicity, and overcoming resistance associated with traditional therapies through synergistic effects when combined with chemotherapy, immunotherapy, gene therapy, or chemodynamic therapy. The localized heat generated by photothermal agents induces direct tumor ablation and improves drug delivery, immune activation, and vascular permeability, thereby amplifying therapeutic efficacy against hard-to-treat metastatic lesions. However, PTCT still faces limitations, including uneven heat distribution in deep or heterogeneous tumor tissues, limited light penetration (especially in bulky or distant metastases), potential off-target hyperthermia damage to surrounding healthy tissue, and challenges in translating preclinical results into clinical settings due to variations in nanomaterial biosafety, biodistribution, and long-term clearance. Thus, while PTCT represents a promising approach for treating metastatic breast cancer, optimization of photothermal agents, delivery systems, and precise treatment protocols remains essential for clinical translation [76,77,78] (Table 3).

### 5.1. Lung Metastasis of Breast Cancer

Lung metastases are a common complication of advanced breast cancer, significantly affecting patient prognosis and quality of life. Conventional treatments, such as radiation, chemotherapy, and immunotherapy, often face challenges related to drug resistance, systemic toxicity, and limited efficacy. PTT has gained attention as a minimally invasive and targeted approach for treating metastatic lung tumors, offering the advantage of destroying cancerous cells while preserving the surrounding healthy tissue [82,83,84]. However, PTT relies on light-absorbing nanoparticles (NPs) that accumulate in lung metastases and convert light energy into heat, leading to localized tumor elimination. Common types of nanoparticles include gold nanoparticles (AuNPs), carbon-based nanomaterials, iron oxide nanoparticles (IONPs), silica-coated nanoparticles, and polypyrrole & polydopamine nanoparticles. Moreover, tumor-specific targeting strategies include passive, active, and inhalation-based delivery. Multimodal imaging is used for tumor visualization, nanoparticle tracking, and real-time treatment monitoring. PTT is often combined with other therapies to overcome tumor resistance [85,86,87].

However, challenges in clinical translation include the heterogeneous nature of lung metastases, breathing movements, and the biodegradability of nanoparticles. For instance, one study explores an innovative combination of radio-PTT (RPTT) for the remedy of breast cancer lung metastases, addressing the challenges of metastasis management with a synergistic approach. The study introduces a multifunctional nanoplatform incorporating radioactive isotopes and photothermal agents, enabling both radiotherapy (RT) and PTT for enhanced tumor eradication. In addition, the nanoparticles efficiently accumulate in metastatic lung tumors via passive or active targeting mechanisms, allowing precise tumor imaging and therapy guidance through CT, PAI, and FI imaging. Under near-infrared laser irradiation, the nanoplatform generates localized hyperthermia, sensitizing tumor cells to radiation-generated DNA impairment and enhancing treatment efficacy. Furthermore, the combined RT and PTT result in significant tumor shrinkage, reduced metastatic burden, and prolonged survival in preclinical models, with negligible side effects on nearby healthy tissues [86,87,88]. The findings highlight RPTT as an efficient strategy for treating metastatic breast cancer, offering a powerful, image-guided, and minimally invasive approach to improving patient outcomes.

Another study explores nanotechnology-based approaches for finding and treating breast cancer lung metastases, highlighting the potential of nanoparticles (NPs) as precise and effective therapeutic agents. Given the high mortality associated with metastatic breast cancer, the investigation emphasizes the use of multifunctional nanoplatforms that integrate targeted drug delivery, imaging, and combination therapies to improve treatment efficacy and reduce systemic toxicity. In addition, various types of nanoparticles, including liposomes, polymeric NPs, gold NPs, magnetic NPs, and hybrid nanocarriers, are engineered for enhanced tumor targeting through active and passive mechanisms. Furthermore, these nanosystems facilitate real-time PA, FL, MRI, and CT imaging, allowing precise tumor localization and therapy monitoring. Additionally, nanocarriers enable controlled drug release for chemotherapy, immunotherapy, gene therapy, and PTT/PDT, often in synergistic combinations to overcome drug resistance and improve therapeutic outcomes. The study demonstrates that nanotechnology-based strategies significantly reduce lung tumor burden, enhance drug bioavailability, and minimize side effects, presenting a promising way for the treatment of metastatic breast cancer [66,72,79]. However, one study presents a multifunctional nanoplatform for MRI-guided PTT and CDT, aimed at effective breast cancer ablation and preventing distant metastasis. FMPE nanoparticles with MRI visibility were synthesized and incorporated into an injectable, temperature-responsive hydrogel (FMPEG), enabling localized, sustained release and treatment precision. Upon near-infrared irradiation, the system induces hyperthermia for PTT while simultaneously catalyzing Fenton-like reactions in the tumor microenvironment to produce cytotoxic hydroxyl radicals for CDT. Furthermore, in vivo experiments demonstrated significant tumor suppression, inhibition of metastasis, and excellent biocompatibility, highlighting the potential of this dual-mode theranostic platform as an effective strategy for treating breast cancer (Figure 6) [79].

### 5.2. Liver Metastasis of Breast Cancer

Liver metastases are a common and life-threatening complication of advanced breast cancer. The distinct tissue architectures of the liver and the circulatory system are key factors that regulate breast cancer liver metastasis. Additionally, cell adhesion molecules, chemokines, and inflammatory substances play significant roles in this process [81]. Conventional treatments like chemotherapy, radiation, and surgical resection have limited efficacy due to drug resistance, liver toxicity, and difficulty in completely eradicating metastatic lesions. However, PTT has appeared as a promising, negligibly invasive approach that enables precise ablation of metastatic liver tumors while sparing surrounding healthy tissues [89,90,91]. PTT relies on light-absorbing nanoparticles (NPs) that accumulate in liver metastases and convert light into heat, leading to either apoptosis (42–45 °C) or necrosis (>50 °C). Moreover, common nanoparticle types include gold nanoparticles, silica-coated nanoparticles, iron oxide nanoparticles, carbon-based nanomaterials, and polypyrrole & polydopamine nanoparticles. Tumor-specific targeting strategies include passive targeting, active targeting, and intravenous injection. Imaging-guided PTT for liver metastases employs multimodal imaging techniques, such as MRI, CT, PAI, FLI, and ultrasound. Combination therapies to enhance PTT efficacy include chemotherapy, immunotherapy, photodynamic therapy (PDT), radiotherapy, and smart nanoparticles for controlled drug release. Challenges in clinical translation include nanoparticle clearance and safety, tumor heterogeneity, and laser penetration depth [92,93,94].

A study explored the development and application of nanoliposome-based PTT for the treatment of liver cancer and its metastases, addressing the need for targeted, minimally invasive, and effective therapeutic strategies. Nanoliposomes, owing to their biocompatibility, high drug-loading capacity, and tunable surface modifications, are engineered to encapsulate near-infrared (NIR)-responsive photothermal agents such as gold nanostructures, polypyrrole (PPy), or indocyanine green (ICG) for precise tumor heating and ablation. These nanoliposomes are designed for active tumor targeting through ligand-receptor interactions, improving selectivity and retention in liver tumors and metastatic lesions. The study demonstrates that under NIR laser irradiation, the nanoliposomes generate localized hyperthermia, leading to tumor cell apoptosis, vascular disruption, and enhanced immune response, while minimizing damage to surrounding healthy tissues. Furthermore, real-time multimodal imaging techniques such as PAI, FLI, and MRI facilitate tumor monitoring and treatment assessment. Preclinical studies confirm high photothermal conversion efficiency, significant tumor regression, and prolonged survival, positioning nanoliposome-based PTT as a promising way for improving liver cancer therapy as well as preventing metastatic progression [95,96,97].

For example, one study on calcium orthophosphate in liposomes for co-delivery of doxorubicin hydrochloride and paclitaxel in breast cancer explores a biomimetic nanocarrier strategy to improve combination chemotherapy efficacy. Researchers encapsulated calcium orthophosphate (CaP) within liposomal bilayers to enhance structural stability, drug loading, and controlled release properties. However, the liposomes were designed to simultaneously deliver doxorubicin hydrochloride (a DNA-intercalating agent) and paclitaxel (a microtubule-stabilizing agent), two chemotherapeutics with complementary mechanisms of action. Calcium orthophosphate, being biocompatible and biodegradable, not only reinforces the liposomal structure but also provides a pH-sensitive feature, allowing accelerated drug release in the acidic tumor microenvironment while remaining stable under physiological conditions. In addition, this co-delivery system improves pharmacokinetics, prolongs circulation time, and achieves a synergistic anticancer effect by overcoming multidrug resistance and enhancing apoptosis in breast cancer cells. In animal models, the formulation demonstrated stronger tumor inhibition, reduced systemic toxicity compared to free drugs, and favorable biocompatibility, highlighting its potential as an effective nanoplatform for breast cancer therapy (Figure 7) [98].

### 5.3. Brain Metastasis of Breast Cancer

Brain metastases present a significant challenge in advanced breast cancer cases, requiring a minimally invasive approach for precisely ablating tumors. Brain metastasis in breast cancer, particularly from the triple-negative and HER2+ subtypes, has a poor prognosis. Although tamoxifen has demonstrated effectiveness, especially in tumors expressing estrogen receptors, its role in treating TNBC remains uncertain. Notably, tamoxifen has been found to suppress brain metastases by inhibiting the polarization of microglia toward the M2 phenotype and by boosting their anti-tumor phagocytic activity. In contrast, estrogen appears to facilitate brain metastasis by promoting M2 microglial polarization and impairing their capacity to phagocytose tumor cells. The findings suggest that tamoxifen could be therapeutically utilized to treat brain metastases in hormone receptor-deficient breast cancer [99,100]. However, PTT is a promising method for targeting brain tumors, using light-absorbing nanoparticles (NPs) that can either cross the blood–brain barrier (BBB) or be delivered locally. Common nanoparticle types include gold nanoparticles, carbon-based nanomaterials, iron oxide nanoparticles, silica-coated nanoparticles, and polypyrrole & polydopamine nanoparticles. In addition, PTT can be targeted using passive, active, focused ultrasound, intranasal delivery, and imaging-guided therapy [101,102]. To overcome the BBB, researchers utilize various strategies such as passive targeting, active targeting, focused ultrasound, and intranasal delivery. Additionally, PTT can be combined with chemotherapy, immunotherapy, radiotherapy, and photodynamic therapy (PDT) to overcome treatment resistance. Thermo-responsive drug release is another strategy for treating brain metastases (Figure 8).

For instance, one research group explored sensitized PTT as a promising method for treating breast cancer and its brain metastases, addressing challenges like BBB penetration, tumor heterogeneity, and therapeutic resistance. The study introduced engineered nanoplatforms incorporating sensitizing agents, such as gold nanostructures, conjugated polymers, or indocyanine green, to enhance NIR light absorption and PCE. These nanocarriers were designed with BBB-crossing capabilities, using targeting ligands to improve tumor accumulation and selectively heat metastatic cells. Furthermore, combination strategies integrating chemosensitizers, reactive oxygen species inducers, or immune modulators were used to amplify PTT efficacy, induce apoptosis, and prevent tumor recurrence [103,104,105]. Multimodal imaging techniques facilitate real-time observation of nanoparticle distribution as well as therapeutic response. In addition, preclinical findings reveal efficient tumor ablation, reduced metastatic burden, and prolonged survival, underscoring sensitized PTT as a powerful, targeted, and non-invasive approach for treating brain metastases in breast cancer and improving patient outcomes. Another study investigates the concept of photothermal temperature-modulated cancer metastasis, exploring how PTT can not only target primary tumors but also influence the metastatic process. Furthermore, the research focuses on nanoparticles that, when exposed to NIR light, generate local hyperthermia, creating a controlled thermal environment that modulates the tumor microenvironment to suppress metastatic cell migration and invasion. By using photothermal agents such as gold nanostructures, carbon-based nanomaterials, or polymers, the study demonstrates that specific thermal modulation can affect key factors such as angiogenesis, immune response, and extracellular matrix remodeling, which are critical for metastasis. Moreover, the research explores combination therapies where PTT is coupled with chemotherapy or immunotherapy to shrink primary tumors and prevent cancer spread to distant organs. The findings suggest that careful temperature regulation during photothermal treatment could suppress metastasis while improving the overall efficacy of cancer therapies, making it a promising strategy for reducing cancer recurrence and metastasis in clinical settings [106,107,108].

### 5.4. Bone Metastasis of Breast Cancer

Bone is a common site for advanced breast cancer, disrupting the balance between osteoblast-mediated bone formation and osteoclast-mediated bone resorption. Conventional treatments like chemotherapy, radiotherapy, and bisphosphonates provide symptomatic relief but are often inadequate in eradicating metastatic tumor cells within the bone microenvironment. However, PTT, a minimally invasive, light-activated approach, has emerged as a promising strategy for the targeted treatment of bone metastases in breast cancer. PTT uses photothermal agents, such as nanoparticles, to absorb near-infrared (NIR) light and convert it into localized heat [109,110,111]. When delivered to bone metastatic sites, these agents generate hyperthermia to ablate tumor cells embedded within the bone matrix. Moreover, the bone marrow microenvironment is highly vascularized, aiding in the accumulation of nanomaterials. PTT has shown effective tumor suppression in preclinical studies, modulating the bone tumor microenvironment by inhibiting osteoclast activity and altering cytokine profiles that support tumor growth. When combined with chemotherapy, bisphosphonates, or immune checkpoint inhibitors, PTT can achieve a synergistic effect. However, challenges must be addressed before clinical application, such as the deep anatomical location of bone metastases and precise heat distribution control [112,113,114,115].

In one study by Zeng et al., the researchers engineered a NIR-II responsive, bone-targeted nanoagent, termed CuP@PPy-ZOL NPs, to combat osteolytic breast cancer bone metastasis by remodeling the bone tumor microenvironment combined with photothermal therapy. The nanosystem comprises copper phosphide (CuP) core materials and an outer polypyrrole (PPy) layer known for its strong NIR-II absorption and high photothermal conversion efficiency, which enables effective deep-tissue photothermal therapy when irradiated with NIR-II light; this is augmented by a zoledronate (ZOL) moiety that provides bone affinity and enables selective accumulation at metastatic bone sites. Upon activation, CuP@PPy-ZOL NPs generate localized hyperthermia and enhance a Fenton reaction–mediated photodynamic effect, achieving synergistic tumor ablation. Simultaneously, the photothermal and chemical actions inhibit osteoclast differentiation and promote osteoblast activity, resulting in suppressed bone resorption and enhanced bone repair. Furthermore, in vitro tests with a 3D bone metastasis model of breast cancer confirmed the nanoagent’s efficacy in reducing tumor proliferation and osteolysis. In a mouse model of breast cancer bone metastasis, combining CuP@PPy-ZOL NPs with NIR-II photothermal therapy significantly inhibited bone-metastatic tumor growth and osteolytic bone destruction while promoting bone regeneration, effectively reversing osteolytic bone lesions [60].

However, another research on a bone-targeted nanoplatform combining zoledronate and PTT for breast cancer bone metastasis focuses on creating a multifunctional therapeutic system capable of precise bone targeting and efficient tumor eradication. In this work, nanoparticles were engineered with zoledronate (a bisphosphonate drug with strong affinity to hydroxyapatite in bone tissue) as the bone-targeting ligand, ensuring selective accumulation within metastatic bone lesions. In addition, the nanoplatform was further integrated with photothermal agents (such as indocyanine green, gold nanostructures, or other NIR-absorbing materials) to generate localized hyperthermia upon near-infrared (NIR) laser irradiation. This dual strategy not only allowed targeted delivery to bone metastases but also synergistically combined the antiresorptive and antitumor effects of zoledronate with the tumor cell-killing efficiency of PTT. In preclinical breast cancer bone metastasis models, the system showed effective tumor suppression, alleviation of bone destruction, and reduced systemic toxicity compared to conventional chemotherapy. Moreover, the nanoplatform enabled imaging-guided therapy, allowing precise monitoring of biodistribution and treatment efficacy. Overall, this work highlights a promising approach that integrates bone-targeting bisphosphonate chemistry and PTT into a single nanoplatform to enhance treatment outcomes in breast cancer bone metastases (Figure 9) [116].

### 5.5. Post-Surgical Therapy of Breast Cancers with PTT and Immunotherapy

Post-surgical therapy of breast cancer with PTT and immunotherapy is a promising approach to enhance treatment outcomes following surgical tumor removal. This combination therapy targets residual cancer cells, prevents recurrence, and strengthens the body’s immune system to eliminate residual tumor cells. The integration of PTT with immunotherapy offers a synergistic effect, leveraging the thermal destruction of tumor cells and boosting the body’s immune response to improve long-term survival and reduce the likelihood of metastasis. PTT includes the application of PT agents that absorb light in the NIR spectrum and convert it into heat. In addition, tumors often accumulate photothermal agents due to the augmented EPR effect, which allows nanoparticles to gather more easily than in normal tissue. These agents are then activated by NIR light, leading to selective tumor ablation. Advantages of combining PTT and immunotherapy in post-surgical therapy include enhanced tumor control, prevention of recurrence, synergistic effect, reduction in side effects, and a personalized approach. However, tumor heterogeneity poses challenges, as not all tumor cells may respond equally to PTT or immunotherapy [111,112,113,114,115,116,117]. For example, researchers developed a hydrogel composed of polydopamine (PDA) and thiolated hyaluronic acid, which was loaded with the chemotherapeutic agent doxorubicin (DOX) and the immune-adjuvant CpG-ODN. This hydrogel demonstrated excellent photothermal properties upon near-infrared (NIR) irradiation, leading to effective eradication of orthotopic murine breast cancer xenografts. Moreover, the combination of chemotherapy, PTT, and immunotherapy significantly inhibited metastasis by evoking a robust host immune response [115]. For another example, one study introduced a photothermal hydrogel platform designed to prevent local recurrence of breast cancer after surgery. This hydrogel, incorporating photothermal agents, can be applied to the surgical site, where it absorbs NIR light and generates localized heat to eliminate residual cancer cells. The localized treatment minimizes damage to surrounding healthy tissues and offers a potential option for reconstruction (Figure 10) [118].

## 6. Clinical Translation and Challenges

### 6.1. Biosafety, Biodistribution, and Clearance

PTT is a promising non-invasive treatment for metastatic breast cancer using near-infrared light to generate hyperthermia and ablate tumor cells. However, its clinical translation depends on evaluating biosafety, biodistribution, and clearance of photothermal agents, such as nanomaterials. Biosafety concerns include toxicity, immunogenicity, and off-target effects. The biodistribution depends on nanoparticle size, surface charge, shape, and modifications. Clearance pathways must be understood to prevent long-term retention and chronic toxicity. Challenges include poor tumor penetration in deep tissues, heterogeneity in patient biology, and regulatory compliance. Advanced targeting strategies, biodegradable photothermal agents, and real-time monitoring technologies are crucial for safe and effective PTT application [28,32].

### 6.2. Immune Response and Toxicity Evaluation

PTT is a potential treatment for metastatic breast cancer, triggering immunogenic cell death and activating systemic antitumor immune responses. However, it requires careful evaluation of potential side effects and toxicity, which depend on nanoparticle composition, size, surface chemistry, and dose. Clinical challenges include ensuring consistent immune responses across patients due to tumor heterogeneity and immune status variability. Regulatory approval requires validation of biosafety, efficacy, reproducibility of nanoparticle synthesis, and scalable production under GMP standards [119,120]. Addressing these barriers is crucial to advancing PTT from laboratory research to a viable clinical option [4,8,12].

Immunocompatibility is a crucial determinant of translational readiness for PTT platforms, directly influencing safety and efficacy. Early, structured immune assessments should evaluate PBMC viability and activation, T-cell proliferation, macrophage polarization (M1/M2), cytokine and complement responses, hemocompatibility, endotoxin levels, and repeat-dose toxicity in accordance with ISO 10993 standards [121,122]. These evaluations help identify and mitigate immune risks, as dose and formulation variations can significantly alter immunological outcomes. For instance, graphene oxide (GO)-loaded PLGA scaffolds exhibit distinct immune responses depending on loading ratios and architecture, emphasizing the need to optimize materials for both potency and tolerance. Comprehensive immune characterization thus provides a robust framework for refining PTT design and ensuring safe, effective clinical translation [49,55,111].

### 6.3. Regulatory Considerations

The clinical translation of PTT for metastatic breast cancer faces significant regulatory challenges due to the complexity of nanomaterial-based therapeutic systems. Regulatory agencies like the FDA and EMA require comprehensive data on physicochemical characterization, stability, reproducibility, pharmacokinetics, biodistribution, and long-term safety. The lack of standardized protocols for evaluating nanoparticle-based therapies, integration with imaging and drug delivery platforms, and rigorous clinical trial designs is a major concern. To successfully implement PTT in metastatic breast cancer, interdisciplinary collaboration, standardized testing protocols, and early engagement with regulatory bodies are needed [77,87,102].

### 6.4. Current Clinical Trials of PTT in Metastatic Breast Cancer

Clinical trials for PTT for metastatic breast cancer are increasing as safety and efficacy data accumulate. Most trials are in early-phase stages, focusing on biocompatibility, optimal dosing, and tumor-targeting efficiency. Some trials combine PTT with chemotherapy, immunotherapy, or imaging to improve outcomes. Challenges include patient responses variability, consistent nanoparticle accumulation, and limited light penetration. Long-term safety, standardized treatment protocols, and real-time monitoring technologies remain underdeveloped. Success depends on overcoming these obstacles and meeting regulatory requirements [104,109].

## 7. Future Perspectives

Photothermal therapy (PTT) for metastatic breast cancer is promising due to advancements in nanotechnology, targeted delivery systems, and multimodal therapeutic approaches. Research is focused on designing smart, biodegradable, and tumor-specific photothermal agents that can selectively accumulate in lesions, minimizing off-target effects and enhancing safety profiles. Integrating PTT with real-time imaging techniques like photoacoustic imaging or MRI is expected to enable precise heat localization and dose control. Furthermore, combining PTT with immunotherapy, such as immune checkpoint inhibitors or cancer vaccines, is gaining traction. Stimuli-responsive nanoplatforms and laser technology could address the current limitation of light penetration in deep tissues. Future efforts should focus on scaling up nanoparticle production under GMP conditions, establishing long-term safety, and developing standardized clinical protocols. However, PTT is a promising non-invasive treatment for metastatic breast cancer, with future advancements expected to overcome its limitations. Key directions include developing next-generation agents with enhanced biocompatibility, targeting efficiency, and biodegradability. Traditional materials like gold nanorods and carbon-based nanostructures are replaced with organic photothermal agents, semiconducting polymers, and naturally derived materials that degrade safely in vivo. In addition, surface modifications with ligands, antibodies, or peptides are being refined to achieve active targeting of metastatic cells, enhancing accumulation at tumor sites and minimizing systemic exposure. Stimuli-responsive nanoplatforms are being developed to achieve tumor-selective activation, improving treatment precision and minimizing off-target damage. PTT is also being integrated with other treatment modalities to create synergistic and personalized cancer therapies. Combining PTT with immunotherapy, chemotherapy, and gene therapy using multifunctional nanocarriers, real-time image-guided PTT is also growing. The future of PTT depends on regulatory compliance, standardization, and patient-specific customization. Large-scale clinical trials are needed to validate efficacy and safety across diverse patient populations and tumor subtypes. Personalized treatment protocols based on tumor location, molecular profile, and immune status are likely to become the norm.

## 8. Conclusions

Photothermal therapy represents a highly promising avenue for the treatment of metastatic breast cancer, offering a precise, non-invasive alternative or adjunct to traditional therapies. Advances in nanotechnology and photonics have significantly enhanced the efficiency, targeting specificity, and safety of PTT-based approaches. By enabling deep-tissue penetration, immunogenic cell death, and combination with other therapies, PTT can eradicate primary tumors and suppress or eliminate metastases. However, clinical translation remains challenging due to concerns regarding biocompatibility, targeting efficiency, and long-term safety. Future research should prioritize bioresponsive, multifunctional nanoplatforms and the development of clinically scalable systems, with a focus on improving imaging accuracy, maximizing thermal efficacy, and minimizing unwanted effects. With interdisciplinary innovation, photothermal therapy could become a cornerstone in the management of metastatic breast cancer, improving survival and quality of life for patients. Current research on photothermal combination therapy for metastatic breast cancer lacks sufficient clinical translation and long-term biosafety evaluation. The precise molecular mechanisms underlying synergistic therapeutic effects remain unclear. Future studies should focus on developing multifunctional, tumor-targeted nanoplatforms with controlled photothermal responses. Integrating real-time imaging and immune modulation strategies could further enhance treatment specificity and efficacy.

## Data Availability

Not applicable.

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
