# Peer review of "Photothermal Combination Therapy for Metastatic Breast Cancer: A New Strategy and Future Perspectives"

_biomedicines, 2025, doi:10.3390/biomedicines13102558_

Round 1
Reviewer 1 Report
Comments and Suggestions for Authors
My comments:
1- The abstract needs to be rewritten (consisting of aim, method, results, and conclusions).
2- Authors should define abbreviations at their first appearance in the text (for example, PDT in the abstract).
3- I did not see any section about the methodology of the work. What is the methodology of the work? There is no information regarding the design and methodology of the work. It would be better to draw a flow chart for designing the search.
4- I believe a more recent published research paper is necessary to cover all relevant studies. Therefore, I recommend adding them to the literature. (some articles such as: doi:10.1016/j.pdpdt.2020.101896, doi:10.1016/j.pdpdt.2020.102061, and doi: 10.3390/diagnostics13050833).
5- In my opinion, there is no need to bring all the figures from previous research work that the authors bring into the manuscript.
6- In the Discussion section, in addition to perspectives, authors need to write limitations and advantages.
7- The manuscript requires a review for the English language and grammar.
Comments on the Quality of English LanguageThe manuscript requires a review for the English language and grammar.
Author Response
Reviewer-1
Comments and Suggestions for Authors
My comments:
1- The abstract needs to be rewritten (consisting of aim, method, results, and conclusions).
Response: According to the reviewer's comment, we have updated the information.
Metastatic breast cancer (MBC) remains one of the most aggressive and fatal malignancies in women, primarily due to tumor heterogeneity, multidrug resistance, and the limitations of conventional therapeutic approaches. Aim: This review aims to evaluate recent advances in nanomaterial-based photothermal therapy (PTT) platforms and their potential in the treatment of metastatic breast cancer. Method: A comprehensive analysis of current literature was conducted to examine how various nanomaterials are engineered for targeted PTT, with particular emphasis on their mechanisms of action, synergistic applications with chemotherapy, immunotherapy, and photodynamic therapy, as well as their capacity to overcome challenges associated with targeting metastatic niches. Results: The findings indicate that nanotechnology-enabled PTT provides spatiotemporal precision, efficient tumor ablation, and reduced systemic toxicity, while significantly enhancing therapeutic outcomes when integrated into multimodal treatment strategies. Recent preclinical studies and early clinical trials further underscore advancements in imaging guidance, thermal efficiency, and site-specific drug delivery; however, issues related to biocompatibility, safety, and large-scale clinical translation remain unresolved. Conclusion: Nanomaterial-assisted PTT holds substantial promise for improving therapeutic efficacy against metastatic breast cancer. Future research should prioritize optimizing imaging resolution, minimizing adverse effects, and addressing translational challenges to accelerate clinical integration and ultimately enhance health outcomes for women. (Page 1)
2- Authors should define abbreviations at their first appearance in the text (for example, PDT in the abstract).
Response: According to the reviewer's comment, we have updated the information.
3- I did not see any section about the methodology of the work. What is the methodology of the work? There is no information regarding the design and methodology of the work. It would be better to draw a flow chart for designing the search.
Response: According to the reviewer's comment, we have updated the information.
Scheme 2. Flow chart information regarding the design and methodology of the work
4- I believe a more recent published research paper is necessary to cover all relevant studies. Therefore, I recommend adding them to the literature. (some articles such as: doi:10.1016/j.pdpdt.2020.101896, doi:10.1016/j.pdpdt.2020.102061, and doi: 10.3390/diagnostics13050833).
Response; The above articles have been used and cited in the manuscript.
5- In my opinion, there is no need to bring all the figures from previous research work that the authors bring into the manuscript.
Response: According to the reviewer's comment, we have been revised.
6- In the Discussion section, in addition to perspectives, authors need to write limitations and advantages.
Response: We updated the information. The above contents have been revised
Photothermal combination therapy (PTCT) for metastatic breast cancer offers significant advantages by enhancing tumor selectivity, minimizing systemic toxicity, and overcoming resistance associated with traditional therapies through synergistic effects when combined with chemotherapy, immunotherapy, gene therapy, or chemodynamic therapy. The localized heat generated by photothermal agents induces direct tumor ablation and improves drug delivery, immune activation, and vascular permeability, thereby amplifying therapeutic efficacy against hard-to-treat metastatic lesions. However, PTCT still faces limitations, including uneven heat distribution in deep or heterogeneous tumor tissues, limited light penetration (especially in bulky or distant metastases), potential off-target hyperthermia damage to surrounding healthy tissue, and challenges in translating preclinical results into clinical settings due to variations in nanomaterial biosafety, biodistribution, and long-term clearance. Thus, while PTCT represents a promising approach for treating metastatic breast cancer, optimization of photothermal agents, delivery systems, and precise treatment protocols remains essential for clinical translation. (page-19)
7- The manuscript requires a review for the English language and grammar.
Response: We have updated the information.
Reviewer 2 Report
Comments and Suggestions for Authors
Dear Authors,
The manuscript entittled “Photothermal Combination Therapy for Metastatic Breast Cancer: A New Strategy and Future Perspectives,” is a well-constructed and timely review that clearly synthesizes recent advances in photothermal combination strategies for metastatic breast cancer. The site-by-site organization is practical, highlights key challenges, and pairs them with appropriate strategies—useful for readers planning studies. I have only a few minor recommendations to improve accuracy and consistency:
- Caption year correction (p. 13). In the caption “Reproduced from ref. [49] Copyright Elsevier 2026,” the year appears to be a typo; please update to 2025.
- Figure caption consistency (Figs. 4 and 5). Please standardize the caption style across all reproduced/adapted figures (e.g., “Reproduced with permission from … Journal, Year”) and confirm that permissions are on file where required.
- Reference duplication. Remove duplicate entries in the references (e.g., Ashkarran et al. listed twice; Yu, Hu & Gao listed twice) and renumber citations accordingly.
Author Response
Reviewer-2
Comments and Suggestions for Authors
Dear Authors,
The manuscript entittled “Photothermal Combination Therapy for Metastatic Breast Cancer: A New Strategy and Future Perspectives,” is a well-constructed and timely review that clearly synthesizes recent advances in photothermal combination strategies for metastatic breast cancer. The site-by-site organization is practical, highlights key challenges, and pairs them with appropriate strategies—useful for readers planning studies. I have only a few minor recommendations to improve accuracy and consistency:
Caption year correction (p. 13). In the caption “Reproduced from ref. [49] Copyright Elsevier 2026,” the year appears to be a typo; please update to 2025.
Response: According to the reviewer's comment, we have updated the information.
Figure caption consistency (Figs. 4 and 5). Please standardize the caption style across all reproduced/adapted figures (e.g., “Reproduced with permission from … Journal, Year”) and confirm that permissions are on file where required.
Response: According to the reviewer's comment, we have updated the information.
Reference duplication. Remove duplicate entries in the references (e.g., Ashkarran et al. listed twice; Yu, Hu & Gao listed twice) and renumber citations accordingly.
Response: According to the reviewer's comment, we have updated the information.
Reviewer 3 Report
Comments and Suggestions for Authors
In the manuscript titled “Photothermal combination therapy for metastatic breast cancer: a new strategy and future perspectives”, Guo and co-authors made a comprehensive review for the treatment of metastatic breast cancer, this manuscript is well written and organized, before its publications, some revisions should be addressed. Here are detailed comments:
- While the introduction provides a good overview of existing literature, the manuscript does not clearly highlight the novel aspects of the study or its clinical relevance. The gap in current knowledge and how this work fills that gap are not sufficiently articulated..
- The manuscript’s current structure is somewhat fragmented. Sections overlap or jump between topics without clear transitions, which makes it difficult for readers to follow the narrative.
- The review relies heavily on text and lacks summary tables or schematic figures, which are critical in review articles to synthesize large amounts of information.
- The conclusion section is brief and mainly repeats earlier content without offering forward-looking insights. A strong review should not only summarize existing knowledge but also identify gaps and suggest future directions for research.
- The quality of some figures is low and detailed information cannot be figured out, hence, higher solution of figures should be added in the manuscript.
Author Response
Reviewer-3
Comments and Suggestions for Authors
In the manuscript titled “Photothermal combination therapy for metastatic breast cancer: a new strategy and future perspectives”, Guo and co-authors made a comprehensive review for the treatment of metastatic breast cancer, this manuscript is well written and organized, before its publications, some revisions should be addressed. Here are detailed comments:
While the introduction provides a good overview of existing literature, the manuscript does not clearly highlight the novel aspects of the study or its clinical relevance. The gap in current knowledge and how this work fills that gap are not sufficiently articulated..
The manuscript’s current structure is somewhat fragmented. Sections overlap or jump between topics without clear transitions, which makes it difficult for readers to follow the narrative.
Response: We have updated the information according to the reviewer's comment.
Photothermal combination therapy represents a novel and clinically relevant approach for treating metastatic breast cancer by integrating localized heat generation with complementary therapeutic modalities such as chemotherapy, immunotherapy, or chemodynamic therapy. Unlike conventional treatments with systemic toxicity and poor selectivity, this strategy enables precise spatiotemporal control of tumor ablation while enhancing drug delivery, immune activation, and inhibition of metastatic spread. The development of advanced photothermal nanomaterials capable of deep tissue NIR-II absorption, real-time imaging guidance, and multifunctional therapeutic action offers a transformative potential to overcome drug resistance and improve patient survival outcomes, making it a promising direction for future clinical translation.
Despite significant progress, research on photothermal combination therapy for metastatic breast cancer still faces several critical gaps, including limited clinical translation, inadequate understanding of synergistic mechanisms, and insufficient evaluation of long-term biosafety and biodistribution. Most current studies remain at the preclinical stage, often using small animal models that do not fully mimic human tumor heterogeneity or metastasis. Moreover, the optimization of photothermal agents for deep tissue penetration, precise thermal control, and immune modulation remains a challenge. Future research should focus on designing biocompatible, multifunctional nanoplatforms with image-guided, stimuli-responsive, and immune-integrated capabilities, supported by comprehensive in vivo and clinical studies to validate therapeutic safety, efficacy, and translational potential. (page-3)
The review relies heavily on text and lacks summary tables or schematic figures, which are critical in review articles to synthesize large amounts of information.
Response: According to the reviewer's comment, we have updated the information.
Table 3: Various types of photothermal agents used in PTT with combinatory therapy in metastatic breast cancer.
|
Phototherapeutic agent |
Machanism |
Target Breast metastasis |
In vitro and in vivo study |
Reference |
|
FMPEG |
PTT+CD |
Liver and breast cancer |
Sustained and long-lasting antitumor effect, attributed to the immunostimulatory properties by producing tumor-associated antigens, suppressing distant tumor and liver metastasis. |
[80] |
|
ICG/Fe₃O₄@PLGA-ZOL |
PTT+CT |
Bone and breast cancer |
Extraordinary antitumor therapeutic effects and that these NPs can be accurately located in the medullary cavity of the tibia to solve problems with deep lesions, such as breast cancer tibial metastasis, showing great potential for cancer theranostics. |
[64] |
|
CuP@PPy-ZOL |
PTT+CDT |
Bone and breast cancer |
Significantly inhibited the tumor growth of breast cancer bone metastases and osteolysis while promoting bone repair to achieve the reversal of osteolytic breast cancer bone metastases. |
[107] |
|
IR780@PLGA NPs |
PTT |
Bone and breast cancer |
Exhibited excellent PTT for bone metastases |
[109] |
|
ICG@Cu2-XSe-ZIF-8 |
PTT+CDT |
Bone and breast cancer |
Effectively suppresses the tumor cells in bone tissue and reduces the erosion of bone tissue via suppressing osteoclastogenesis. |
[106] |
The conclusion section is brief and mainly repeats earlier content without offering forward-looking insights. A strong review should not only summarize existing knowledge but also identify gaps and suggest future directions for research.
Response: According to the reviewer's comment, we have updated the information.
Current research on photothermal combination therapy for metastatic breast cancer lacks sufficient clinical translation and long-term biosafety evaluation. The precise molecular mechanisms underlying synergistic therapeutic effects remain unclear. Future studies should focus on developing multifunctional, tumor-targeted nanoplatforms with controlled photothermal responses. Integrating real-time imaging and immune modulation strategies could further enhance treatment specificity and efficacy. (Page-40)
The quality of some figures is low and detailed information cannot be figured out, hence, higher resolution of figures should be added in the manuscript.
Response: We have updated the information according to the reviewer's comment.
Reviewer 4 Report
Comments and Suggestions for Authors
The manuscript offers a broad and timely overview of photothermal therapy (PTT) and combination strategies for metastatic breast cancer (MBC), spanning mechanisms, targeting across metastatic sites (lung, liver, brain, bone), and translational challenges. The clinical-translation perspective aligns with the stated aims. Below are my comments:
- Please make the review process explicit: databases searched, date ranges, search strings, and data-extraction strategy.
- Separate mechanisms more clearly: delineate PTT’s non-radiative heat generation from PDT’s ROS-mediated cytotoxicity, and relocate PDT-specific photophysics (e.g., intersystem crossing, singlet-oxygen generation, Jablonski diagram) to a dedicated PDT or PTT+PDT combination subsection to avoid conceptual drift. For PTT, please report temperature–time profiles and thermal dose metrics (e.g., CEM43) alongside illumination parameters; for PDT, report ROS metrics (e.g., singlet-oxygen assays) and oxygen dependence to maintain mechanistic clarity.
- Standardize in vivo reporting with a concise checklist (table) capturing wavelength, irradiance/power density, spot size, exposure time/duty cycle, and thermal dose with typical ranges (e.g., CEM43 or complete temperature–time profiles), plus adverse events. Add a simple schematic mapping NIR-I vs NIR-II suitability by anatomical site. This will enable cross-study comparison and facilitate clinical translation.
- Immunocompatibility represents a foundational pillar of translational readiness. Beyond general biosafety, platforms intended for PTT should undergo an early, structured immune assessment—covering PBMC viability/activation, T-cell proliferation, macrophage polarization (M1/M2), cytokine/complement profiling, hemocompatibility, endotoxin testing, and repeat-dose systemic toxicity—aligned with ISO 10993 families. Dose- and formulation-dependent immune effects reported, for example, for GO-loaded PLGA scaffolds (PMID: 39232353) underscore the need to optimize loading ratios and scaffold architecture before advancing toward clinical use.
- Differentiate preclinical from early-phase clinical evidence and temper translational claims with optical-penetration constraints (NIR-I vs NIR-II), anatomical accessibility, and the role of interventional delivery (probe- or needle-based illumination) for deep lesions.
- Add a compact comparative table contrasting PTT alone vs combination regimens (chemo, immunotherapy, PDT, radiotherapy), distinguishing “mild-PTT” immunomodulatory protocols from ablative protocols, and summarizing typical parameter ranges and adverse events.
Minor comments
- Reformat this sentence below the authors “Zun Wang and Ikram Hasan contributed equally to this manuscript.”
- Reduce overlap between PTT heating physics and PDT-oriented Jablonski/ROS content; keep ROS mechanisms within the PTT+PDT combination subsection.
- Normalize units (irradiance in W·cm⁻², temperature in °C) and consistently label spectral windows (NIR-I, NIR-II).
Author Response
Reviewer-4
Comments and Suggestions for Authors
The manuscript offers a broad and timely overview of photothermal therapy (PTT) and combination strategies for metastatic breast cancer (MBC), spanning mechanisms, targeting across metastatic sites (lung, liver, brain, bone), and translational challenges. The clinical-translation perspective aligns with the stated aims. Below are my comments:
Please make the review process explicit: databases searched, date ranges, search strings, and data-extraction strategy.
Response: This review was designed by systematically searching electronic databases, including PubMed, Web of Science, Scopus, Embase, and Google Scholar for studies published between 2010 and 2025 using the keywords “photothermal therapy,” “combination therapy,” “synergistic therapy,” and “metastatic breast cancer.” Only peer-reviewed English articles that specifically focused on photothermal combination therapies for metastatic breast cancer were included in the review. Screening was conducted through title and abstract evaluation, followed by full-text review to determine relevance, and data were extracted on photothermal agents, therapeutic combinations, imaging modalities, and preclinical or clinical outcomes. The selected articles were then synthesized and thematically organized to highlight strategies such as PTT combined with chemotherapy, immunotherapy, chemodynamic therapy, and other emerging modalities, with particular emphasis on therapeutic efficacy and future perspectives. (Page-5)
Separate mechanisms more clearly: delineate PTT’s non-radiative heat generation from PDT’s ROS-mediated cytotoxicity, and relocate PDT-specific photophysics (e.g., intersystem crossing, singlet-oxygen generation, Jablonski diagram) to a dedicated PDT or PTT+PDT combination subsection to avoid conceptual drift. For PTT, please report temperature–time profiles and thermal dose metrics (e.g., CEM43) alongside illumination parameters; for PDT, report ROS metrics (e.g., singlet-oxygen assays) and oxygen dependence to maintain mechanistic clarity.
Response: We have updated the information according to the reviewer's comment.
Figure 2. (A) PTT agents, when irradiated with a light source, absorb radiation and convert it to heat energy, resulting in cell death. (B) PDT operates through two key reactions upon light absorption by a photosensitizer: a type-I reaction generating radicals and a type-II reaction producing oxidative substrates, both leading to cell damage [36,37].
PTT and PDT are two light-based cancer treatments that differ fundamentally in their mechanisms. PTT relies on the non-radiative relaxation of photoexcited electrons in photothermal agents, such as gold nanorods, carbon-based nanomaterials, or semiconducting polymers, to generate localized heat, inducing protein denaturation, membrane disruption, and apoptosis or necrosis in an oxygen-independent manner, making it effective even in hypoxic tumors. In contrast, PDT depends on photoactivated photosensitizers transferring energy to molecular oxygen to produce reactive oxygen species (ROS), including singlet oxygen and free radicals, which cause oxidative damage to lipids, proteins, and DNA, resulting in apoptosis or necrosis; this process is oxygen-dependent, limiting its efficacy in hypoxic regions. While PTT’s cytotoxicity is primarily thermal, PDT’s is oxidative, and modern nanoplatforms often combine both approaches to exploit the synergistic effects of heat and ROS, enhancing therapeutic outcomes and overcoming the individual limitations of each modality.
For PTT, it is crucial to describe temperature time profiles and quantitative thermal dose metrics, such as the cumulative equivalent minutes at 43 °C (CEM43), to ensure accurate evaluation of therapeutic efficacy and reproducibility. Detailed illumination parameters, including laser wavelength, irradiance (mW·cm⁻²), exposure duration, and total fluence (J·cm⁻²), should be fully described, along with spot size and beam uniformity. Real-time temperature monitoring using thermal cameras, fiber-optic probes, or infrared thermometry should capture spatial and temporal heating behavior, both in vitro and in vivo. Reporting thermal dose allows comparison across studies and distinguishes photothermal effects from other mechanisms such as photodynamic or photochemical processes. Together, standardized thermal profiling and illumination documentation provide the mechanistic transparency necessary to interpret PTT outcomes and optimize therapeutic safety and precision.
To maintain mechanistic clarity in PDT, it is essential to quantitatively describe reactive oxygen species (ROS) generation, particularly singlet oxygen (¹O₂), and assess oxygen dependence under well-defined irradiation conditions. Reliable assays such as Singlet Oxygen Sensor Green (SOSG), DPBF bleaching, EPR spin trapping, or direct near-infrared (1270 nm) phosphorescence should be used to confirm ¹O₂ formation, while general ROS indicators (e.g., DCFH-DA) can complement species-specific results. Experimental details, including light wavelength, irradiance, fluence, photosensitizer concentration, and oxygen levels (normoxia versus hypoxia) must be clearly documented. Comparative studies under different oxygen conditions and with ¹O₂ quenchers (e.g., sodium azide) help confirm oxygen-dependent mechanisms. Reporting quantitative parameters such as singlet oxygen quantum yield (ΦΔ), ROS generation rates, and oxygen consumption ensures reproducibility and strengthens mechanistic interpretation. These practices collectively enable accurate evaluation of PDT efficacy and its dependence on oxygen-mediated photochemical pathways.
Standardize in vivo reporting with a concise checklist (table) capturing wavelength, irradiance/power density, spot size, exposure time/duty cycle, and thermal dose with typical ranges (e.g., CEM43 or complete temperature–time profiles), plus adverse events. Add a simple schematic mapping NIR-I vs NIR-II suitability by anatomical site. Here’s your NIR-I vs NIR-II suitability mapping converted into a clear, publication-ready table format:
Response: We have updated the information according to the reviewer's comment.
Table 1: NIR-I vs NIR-II suitability by anatomical site
|
Depth (mm) |
Typical Anatomical Examples |
Best Spectral Window (General) |
Reference |
|
0 – 3 |
Skin surface, superficial lesions, subcutaneous tissue |
NIR-I (700–900 nm), good contrast, widely available lasers/dyes |
[118] |
|
3 – 10 |
Subcutaneous tumors, small nodal lesions |
NIR-I → NIR-IIa (1000–1350 nm), NIR-II improves depth & contrast |
[119] |
|
10 – 30 |
Deep-seated tumors (organ surface), muscle |
NIR-IIa/b (1000–1350 / 1500–1700 nm), preferred for penetration |
[120] |
|
>30 |
Large organs, brain (through craniotomy), and surgical guidance |
NIR-II (esp. 1000–1350 nm), and interstitial delivery (fibers) recommended |
[121] |
Immunocompatibility represents a foundational pillar of translational readiness. Beyond general biosafety, platforms intended for PTT should undergo an early, structured immune assessment—covering PBMC viability/activation, T-cell proliferation, macrophage polarization (M1/M2), cytokine/complement profiling, hemocompatibility, endotoxin testing, and repeat-dose systemic toxicity—aligned with ISO 10993 families. Dose- and formulation-dependent immune effects reported, for example, for GO-loaded PLGA scaffolds (PMID: 39232353) underscore the need to optimize loading ratios and scaffold architecture before advancing toward clinical use.
Response: We have updated the information according to the reviewer's comment.
Immunocompatibility is a crucial determinant of translational readiness for PTT platforms, directly influencing safety and efficacy. Early, structured immune assessments should evaluate PBMC viability and activation, T-cell proliferation, macrophage polarization (M1/M2), cytokine and complement responses, hemocompatibility, endotoxin levels, and repeat-dose toxicity in accordance with ISO 10993 standards. These evaluations help identify and mitigate immune risks, as dose and formulation variations can significantly alter immunological outcomes. For instance, graphene oxide (GO)-loaded PLGA scaffolds exhibit distinct immune responses depending on loading ratios and architecture, emphasizing the need to optimize materials for both potency and tolerance. Comprehensive immune characterization thus provides a robust framework for refining PTT design and ensuring safe, effective clinical translation [49, 55, 111, 119]. (Page-38)
Differentiate preclinical from early-phase clinical evidence and temper translational claims with optical-penetration constraints (NIR-I vs NIR-II), anatomical accessibility, and the role of interventional delivery (probe- or needle-based illumination) for deep lesions.
Response: We have updated the information according to the reviewer's comment.
Preclinical studies using small-animal models provide controlled settings for testing optical imaging probes and therapies, offering insights and proof of concept. However, they may overestimate translational feasibility due to differences in tissue characteristics between animals and humans. Early-phase clinical evidence, while limited, is crucial for assessing safety and preliminary efficacy in humans. Key considerations include optical penetration differences between NIR-I and NIR-II wavelengths, anatomical accessibility issues for deep lesions, and complexities associated with interventional delivery methods. Aligning translational expectations with these factors can prevent overinterpretation of preclinical findings. (page-8)
Add a compact comparative table contrasting PTT alone vs combination regimens (chemo, immunotherapy, PDT, radiotherapy), distinguishing “mild-PTT” immunomodulatory protocols from ablative protocols, and summarizing typical parameter ranges and adverse events.
Response: We have updated the information according to the reviewer's comment.
Table 2: PTT alone vs combination regimens and parameter ranges
|
Feature |
Mechanism |
Temperature Range (°C) |
Duration |
Typical Adverse Events |
Mechanistic Notes |
Preclinical Stage |
Reference |
|
PTT Alone – Mild (Immunomodulatory) |
Immune activation, ICD |
41–45 |
5–20 min |
Mild inflammation, transient edema |
ICD, HSP release, T-cell priming |
Mostly preclinical |
[123] |
|
PTT Alone – Ablative |
Tumor ablation |
50–60+ |
5–10 min |
Tissue necrosis, pain |
Direct cytotoxicity |
preclinical |
[124] |
|
PTT + Chemotherapy |
Synergistic cytotoxicity |
45–55 |
5–20 min |
Systemic chemo toxicity |
Drug release/heat synergy |
Preclinical |
[125] |
|
PTT + Immunotherapy |
Enhanced anti-tumor immunity |
41–45 |
5–20 min |
Immune-related events |
T-cell activation enhance |
Preclinical |
[126] |
|
PTT + PDT |
ROS-mediated cytotoxicity |
45–55 |
5–20 min |
Skin/photosensitivity, inflammation |
ROS + hyperthermia synergy |
Preclinical |
[127] |
|
PTT + Radiotherapy |
Radiosensitization, cytotoxic synergy |
45–55 |
5–20 min |
Radiation dermatitis, tissue injury |
Hyperthermia enhances DNA damage |
Preclinical |
[128] |
Minor comments
Reformat this sentence below the authors “Zun Wang and Ikram Hasan contributed equally to this manuscript.”
Response: We have updated the information according to the reviewer's comment.
Reduce overlap between PTT heating physics and PDT-oriented Jablonski/ROS content; keep ROS mechanisms within the PTT+PDT combination subsection.
Response: According to the reviewer's comment, we have updated the information.
Normalize units (irradiance in W·cm⁻², temperature in °C) and consistently label spectral windows (NIR-I, NIR-II).
Response: According to the reviewer's comment, we have updated the information.
Round 2
Reviewer 3 Report
Comments and Suggestions for Authors
Authors have addressed all my concern and current manuscript is suitable for publication.